# Strength training among professional UCI road cyclists: Practices, challenges, and rationales

Vidar Vikestad, Idar Kristian Lyngstad, Terje Dalen *

Department of Physical Education and Sport Science, Faculty of Education and Arts, Nord University, Levanger, Norway

* terje.dalen@nord.no

## Abstract

In our study, we aimed to investigate professional road cyclists' practices in strength training, as well as their challenges with and rationales for strength training. Employing a mixed-methods approach, we collected data using a quantitative questionnaire and analysed semi-structured interviews using content analysis. After identifying professional road cyclists on FirstCycling.com's cycling statistics database, we contacted 624 cyclists by direct messaging on social media platforms, primarily Instagram. Of them, 147 professional road cyclists—88 male and 59 female—responded to the questionnaire, which we complemented with 10 semi-structured interviews to gain deeper insights into their strength training. Results revealed a common practice of engaging in two strength training sessions per week during the off- and pre-season, which dropped to one maintenance session during the race season. More female than male cyclists reported engaging in strength training during both the pre-season (96.6% vs. 85.3%) and race season (66.1% vs. 36.4%). The most significant challenges to maintaining consistent strength training were travelling, fatigue, and racing, while the primary rationales for strength training were to enhance performance, prevent injuries, and improve bone health. Moreover, the cyclists' choices in strength training seemed to be largely influenced by guidance from coaches. Altogether, our findings highlight the complexities inherent in strength training among professional cyclists and emphasise the need for expert guidance to maintain consistent practices in strength training, especially during the race season.

## Introduction

Professional road cycling, a demanding endurance sport with varied races and multi-week stage races, drives athletes and coaches to seek competitive advantages via different approaches to training. Several studies have indeed shown that strength training can enhance cycling and endurance performance [1–7]. For improved exercise economy and performance, two weekly pre-season sessions focused on

**Data availability statement:** All relevant data are within the manuscript and its Supporting Information files.

**Funding:** The author(s) received no specific funding for this work.

**Competing interests:** The authors have declared that no competing interests exist.

maximal strength are often recommended [2,3,5,8,9]. Improvements in performance in particular are often attributed to improved exercise economy, thought to be caused by adaptations in the strength-trained muscles, including the postponed activation of less efficient type II fibres, improved neuromuscular efficiency, the conversion of fast-twitch type IIX fibres into more fatigue-resistant type IIA fibres, and improved musculotendinous stiffness [1,2,5]. Those adaptations are likely to disappear if cyclists cease to perform strength training for more than 8 weeks but can be maintained with as little as one session per week [10,11]. Moreover, the improvements in endurance performance due to strength training appear to be equally beneficial for male and female cyclists [6].

A popular form of training among cyclists is core and stability training. Despite having no universal definition, core stability is often limited to the spinal extensors, flexors, lateral flexors, and rotators. However, in this article, *core and stability training* refers to exercises targeting muscles between the shoulders and pelvis [12]. Although core and stability training is widely practised, evidence of its impact on performance remains limited. In a study with trained cyclists, Sitko et al. [13] found that 12 weeks of core exercises encompassing glute bridges, abdominal planks, and prone back extensions did not improve mean power output across various maximal cycling efforts lasting from 5 seconds to 20 minutes. By contrast, their study's group of cyclists who underwent conventional strength exercises (i.e., half squats, single-leg leg presses, one-legged hip flexion, and ankle plantar flexion) showed improvements in time trials [13]. Albeit hypothesised that core training might also boost injury prevention [14], some research suggests that classical strength exercises are sufficient for adaptations in core and stability [12]. After all, lower back pain and knee pain are the most common injuries due to overuse in cycling [15]. Abt et al. [16] have suggested that altered cycling mechanics due to induced core fatigue, as observed among competitive cyclists in their study, could elevate the risk of injury due to subjecting the knee joint to increased stress. Even so, strength training can also provide an important stimulus for maintaining optimal bone health, and that benefit might be particularly important for professional cyclists, who have shown lower bone mineral density and bone mineral content than amateur cyclists [17,18]. Added to that, cyclists often prioritise a low body weight, which could negatively affect their bone health if they do not manage to meet their personal energy demands [19,20]. Bone health might additionally be important for preventing injuries, considering that fractures are the most common acute injury from crashes that have led to withdrawal from the Tour de France [21].

Although strength training can enhance endurance performance, concurrent endurance and strength training can inhibit the development of both strength and hypertrophy [22–25]. Strength training can also temporarily impair endurance performance if recovery between a strength training session and a subsequent endurance session is inadequate [9,26–28]. Impairment in endurance performance following strength training is thought to be caused by neuromuscular fatigue, muscle soreness, and muscle glycogen depletion, among other factors [29], with a magnitude that seems to vary based on the athlete's background in strength training [30]. To

minimise such interference, researchers recommend scheduling strength training and endurance sessions at least 6 hours apart [28,31,32].

For endurance athletes, integrating strength training into training routines can present numerous challenges. A study investigating challenges related to strength training among amateur long-distance triathletes, for instance, revealed time constraints and lack of knowledge about strength training as major barriers [33]. However, professional road cycling stands out among endurance sports for its frequent multi-day stage races, including three Grand Tours lasting 3 weeks each. Moreover, during the 2023 season, the top 400 most frequently competing professional male cyclists averaged 70.8±6.1 race days, while the top 50 female cyclists averaged 53.5±3.8 [34]. Such an extensive racing schedule may introduce challenges with maintaining a consistent strength training routine. In fact, when Gallo et al. [35] analysed the training of three of the top five finishers in the Giro d'Italia in the 22 weeks preceding the event, they found that none of the cyclists were performing strength training despite their coaches' indications to add it to their routines. In a similar study conducted a year later, two of the top five finishers of the Tour de France reported engaging in strength training during November and December but not during the last 6 months preceding the event in July [36]. Nevertheless, no information is provided on why the cyclists discontinued their strength training when approaching the race season.

Although several other studies have investigated the training habits of elite and professional road cyclists, they have often been case studies or multiple-case studies [35–37]. Meanwhile, other studies have examined cyclists' endurance training [38–42] and racing demands [43] but not their habits in strength training. Despite the insights afforded by such research, no studies have investigated the strength training practices of elite or professional road cyclists, the challenges that they encounter, or the rationales behind their choices when it comes to strength training. Given the distinct racing and training demands of road cycling and the complexities surrounding strength training in the sport, in our study we aimed to answer three questions, as follows, while investigating potential differences between male and female cyclists:

1. How do professional road cyclists integrate strength training into their training routines?

2. What challenges related to strength training do they experience?

3. What is the rationale behind their decision-making related to strength training?

## Method

### Participants

Our study's sample included only road cyclists from professional teams categorised by the Union Cycliste Internationale (UCI) as Women's World Teams (WTW), Women's Continental Teams (CTW), World Teams (WTT), Pro Teams (PRT), or Continental Teams (CTM). Of 624 professional road cyclists initially contacted, 147 cyclists—88 male and 59 female—responded to our questionnaire, for a response rate of 23.6%. Our semi-structured interviewees, meanwhile, consisted of 10 cyclists—7 male and 3 female—across all five categories of professional teams.

We identified prospective participants on FirstCycling.com, a cycling statistics database that often provides links to the riders' social media profiles (e.g., on Instagram), and contacted them via direct messaging. Sampling did not exclude any teams or cyclists from the WTW, WTT, or PRT categories, for those teams typically have extensive racing calendars centred around European and international events. However, for the CTW and CTM teams, sampling targeted European-based teams and cyclists as well as non-European teams competing on a primarily European racing calendar. The approach ensured a more homogeneous sample in terms of race exposure and competition structure, for many non-European teams often have very small or regionally focused racing calendars. Ultimately, recruitment was biased towards cyclists with links to their social media profiles on FirstCycling.com, cyclists active on social media who saw our call for participants, and cyclists willing to complete the questionnaire. For the interviews, we contacted cyclists who responded to our initial request to complete the questionnaire, meaning that recruitment for the interview entailed the same selection

biases as recruitment for the questionnaire. Initially, the approach yielded only male interviewees; thus, to ensure representation across all team levels and sexes, we selectively contacted only female cyclists after a sufficient number of male participants were recruited.

## Informed consent and ethics

Participants received a detailed explanation of our research, its procedures, and their rights via an informed consent form. Participation was voluntary, and withdrawal was allowed at any time. Participants' written informed consent was obtained electronically via digital signature prior to the interviews, and their confidentiality was maintained by removing personal identifiers from transcripts and using numbers and team levels for identification. All data were securely stored with password protection until the project's completion and then deleted. The study was reviewed and approved by the Norwegian Agency for Shared Services in Education and Research (Ref. No. 322042) (Table 1).

## Research design

Our study followed a mixed-methods approach combining a questionnaire and semi-structured interviews to investigate the practices, challenges, and rationales of professional road cyclists when it comes to strength training. The questionnaire provided quantitative data for statistical analysis, while interviews offered qualitative insights, for an altogether comprehensive understanding of the topic.

## Questionnaire

We collected data for our study primarily using a questionnaire designed to efficiently gather data on the practices, challenges, and rationales of cyclists in relation to strength training (S1 Fig). All data, from both the questionnaire and interviews, were collected between 11 November 2023 and 20 December 2023, a period corresponding to professional cyclists' off-season or pre-season. Self-reported training data from elite endurance athletes have been found to be

**Table 1. Descriptive information on the distribution of sex, team level, and reported race days among questionnaire respondents, along with the interview number, sex, team level, and label of each interviewee.**

**Questionnaire participants**

| Sex | n | | Age M + SD | Race days M + SD |
|---|---|---|---|---|
| ♀Female | 59 | | 26.3 ± 4.5 | 39.3 ± 15.5 |
| ♂Male | 88 | | 24.6 ± 3.9 | 46.2 ± 15.5 |
| **Team level↓** | | | | |
| ♀WTW | 24 | | 27.3 ± 4.2 | 43.3 ± 12.8 |
| ♀CTW | 35 | | 25.5 ± 4.6 | 36.5 ± 16.7 |
| ♂WTT | 15 | | 27.3 ± 4.4 | 54.1 ± 14.3 |
| ♂PRT | 21 | | 25.6 ± 3.5 | 47.1 ± 18.7 |
| ♂CTM | 52 | | 23.5 ± 3.4 | 43.6 ± 13.8 |
| Total | 147 | | 26.3 ± 4.2 | 43.3 ± 15.8 |

**Interview participants**

| Interviewee | 1♂ | 2♂ | 3♂ | 4♂ | 5♂ | 6♂ | 7♀ | 8♂ | 9♀ | 10♀ |
|---|---|---|---|---|---|---|---|---|---|---|
| Team level | PRT | CTM | CTM | PRT | CTM | PRT | CTW | WTT | WTW | WTW |
| Label | PRT1 | CTM2 | CTM3 | PRT4 | CTM5 | PRT6 | CTW7 | WTT8 | WTW9 | WTW10 |

Abbreviations: ♂ = male, ♀ = female, *SD* = standard deviation, CTM = Continental Team, PRT = Pro Team, WTT = World Team, CTW = Women's Continental Team, and WTW = Women's World Team.

accurate [44]. The questionnaire, tested and refined in a pilot study with amateur cyclists before being distributed to professional cyclists, was administered in English because no translation was deemed necessary for the targeted population.

The questionnaire began by gathering the professional road cyclists' demographic information (i.e., age, sex, and team level), number of race days, and the duration of effort at which they considered themselves to be strongest. Another question inquired into their frequency of strength training, quantified as sessions per week during the off-season, pre-season, and race season with nine response options (i.e., 0, < 1, 1,... 7 sessions per week). When the average number of sessions per week was used in analysis, responses of <1 session per week were counted as 0.5 sessions per week. Five other questions, all answered on a 9-point Likert scale, addressed contentment with coaching received for (1) endurance training and (2) strength training, (3) enjoyment of endurance training, (4) enjoyment of strength training, and (5) confidence in the idea that strength training could improve their cycling performance. Cyclists were also asked whether they had experienced improvements in cycling performance due to strength training, with the response options of 'Yes', 'No', and 'Not sure'. Last, six additional single-choice and multiple-choice questions with preset alternatives addressed (1) methods of strength training, (2) body parts regularly trained, (3) reasons for engaging in strength training, (4) challenges in strength training, and beliefs regarding strength training's (5) positive effects and (6) negative effects. Throughout the questionnaire, we emphasised neutral questions in order to ensure unbiased responses, thereby enhancing the survey's reliability and precision.

## Quantitative analysis

We analysed data from the questionnaire using IBM SPSS Statistics (version 28). Between-group differences in categorical variables (e.g., challenges with strength training) were analysed using the chi-square ($\chi^2$) test of independence, with Cramer's $V$ for effect size (interpreted as small: $V \geq .1$, moderate: $V \geq .3$, or large: $V \geq .5$). To examine within-group differences in categorical data, we employed McNemar's pairwise comparison with Bonferroni correction to adjust for multiple comparisons.

For the questions regarding training frequency and scale-based questions with ordinal data, Mann–Whitney $U$ tests were used to assess between-group differences. To assess the magnitude of between-group differences, we calculated effect sizes ($r$) for each Mann–Whitney $U$ test using the formula:

$$r = \frac{|Z|}{\sqrt{N}}$$

in which $Z$ is the test statistic and $N$ being the total sample size. Results were interpreted to be small ($r \geq .1$), moderate ($r \geq .3$), or large ($r \geq .5$).

Last, for non-normal distributed scale-based questions, Wilcoxon signed-rank tests were used to assess within-group differences in the frequency of strength training across different periods of the season.

## Interviews

Interviews were conducted digitally via Microsoft Teams and recorded for transcription. Two semi-structured interview guides were used: one for cyclists who included strength training in their routine, and one for those who did not. Both guides followed a consistent structure, asking about training routines, challenges, and rationale regarding strength training, with open-ended questions to encourage detailed responses. The interview guide was based on the questionnaire to delve deeper into the research questions and avoid bias, which is crucial in semi-structured interviews [45]. All interviews were conducted by the first author, a former elite road cyclist, who aimed to provide in-depth insights while being mindful of potential biases. The duration of the interviews varied from 15 to 60 minutes depending on the depth of the participant's responses. Efforts to mitigate those biases included avoiding leading questions, actively listening, and critically reflecting on personal biases throughout the interview process.

## Qualitative analysis

A content analysis [45,46] was performed on the interviews, summarised in Table 2, which outlines research questions, derived codes, and their essence. For example, 'Exercises and execution' details how strength training routines are performed. Such data provides insights into participants' practices, challenges, and rationale.

After transcribing the interviews, we conducted a deductive content analysis based on our three research topics of practices, challenges, and rationale in relation to strength training. Codes were developed from the results of the questionnaire and themes identified in the interviews, and transcripts were read multiple times to ensure accurate coding. Individual tables were created for each research topic to organise coded data for analysis and reporting. Coded data were synthesised to identify key insights, and summaries with selected citations to support findings were written. The anonymisation of cyclists, with their names and team identifiers systematically removed, ensured the confidentiality of all participants' data.

# Results

## Characteristics of cyclists

The Chi-square test revealed no significant differences between male and female cyclists, and the McNemar test showed no within-group differences in the duration of effort at which cyclists considered themselves strongest after Bonferroni correction for multiple comparisons. These findings indicate a balanced representation of rider characteristics in both groups.

**Table 2. Overview of the content analysis of data from the interviews.**

| Research topic | Codes | Essence of codes | Participants with the codes |
|---|---|---|---|
| Strength training practice | Exercises and execution | How strength training was performed, which exercises, and how they were performed | 2, 3, 4, 5, 6, 7, 8, 9, 10 |
| | Race season strength training adjustments | What adjustments are made to strength training from pre-season to race season | 2, 3, 6, 7, 8, 9 |
| | Timing of strength training | Where in the training schedule strength training is performed | 2, 3, 4, 5, 6, 7, 8, 9, 10 |
| Challenges with strength training | Fatigue | Fatigue from strength training negatively affects cycling training and performance. | All |
| | Travelling | Travelling makes it difficult to perform strength training consistently. | 1, 2, 4, 5, 6, 10 |
| | Racing | The racing schedule makes it difficult to perform strength training consistently. | 1, 2, 3, 4, 5, 6, 9, 10 |
| | Facilities | How access to training facilities affects strength training | 2, 3, 5, 7, 8, 10 |
| | Enjoyment or motivation | Enjoyment with and motivation for strength training | 1, 5, 8, |
| Rationale for strength training choices | Cycling performance | How strength training can improve cycling performance | 1, 2, 9, 10 |
| | Perceived effect | Perceived effect of strength training | 2, 3, 4, 6, 8, |
| | Risk of injury | To reduce the risk of injury | 2, |
| | Bone health | To improve bone health | 7, 8, 9, 10 |
| | General health | To improve general health | 8, 9, 10 |
| | Coaching guidance | Influence of guidance from coaches on strength training practices | 1, 2, 3, 4, 5, 6, 7, 9, 10 |
| | Stopped strength training during race season | Why strength training is not prioritised during the race season | 4, 5 |
| | Does not engage in strength training at all | Why strength training is not prioritised at any part of the season | 1 |

## Frequency of strength training

As shown in Table 3, more cyclists engaged in strength training (i.e., ≥ 1 session/week) during the off-season (76.2%) and pre-season (89.8%) than during the race season (48.3%). The values for the pre-season were also higher than for the off-season. By sex, female cyclists engaged in strength training (≥1 session/week) more than male cyclists during the pre-season (96.6% vs. 85.3%) and race season (66.1% vs. 36.4%), as well as engaged in it more often than male cyclists during both seasons (pre-season: 2.1 vs. 1.9 session/week, $p < .01$; race season: 1.2 vs. 0.9 sessions/week, $p < .01$). Although statistically significant, the difference of 0.2 and 0.3 sessions/week between sexes in the pre-season and race season represents a small effect size (pre-season: r = .260, and race season: r = .294), suggesting a modest but consistent trend in greater engagement among female cyclists.

## Methods of strength training

As Table 4 details, the most frequently reported methods of strength training were core training and maximal strength training, followed by explosive strength training and hypertrophy training. For both sexes, the lower body and core were the body parts most trained. However, significantly more female than male cyclists reported engaging in upper body strength training (27.1% vs. 8.0%, $p < 0.01$), though the effect size was small (V = .259). Similarly, female cyclists more frequently performed hypertrophy training (55.9% vs. 31.8%, $p < 0.01$, V = .240) and core and stability training (86.4% vs. 71.6%, $p < 0.05$, V = .174), but again, effect sizes were small, suggesting these differences, while statistically significant, were modest in magnitude.

During interviews, cyclists provided insights into their strength training practices, which typically consisted of two weekly sessions during the off-season and pre-season, each featuring three to five primary exercises targeting the lower body muscles used in cycling. Common exercises included squats, single-leg squat variations, leg press, and deadlifts, and core training was often supplementary. In an adaptation phase at the beginning of the off- or pre-season, strength training featured more repetitions and less intensity, subsequently followed by a phase featuring increased intensity and fewer repetitions. Cyclists often focused on an explosive concentric part of the lift and adapted strength exercises to better match the demands of cycling. As one participant stated:

> "Yes, it's mostly two strength sessions per week, typically on the same day as a high-intensity workout. We start with a conditioning phase with 10–12 repetitions, then gradually decrease after 4–5 weeks, down to eight repetitions, and

**Table 3.** Mann–Whitney *U* test comparing differences in sessions per week between male and female cyclists, and a Wilcoxon signed rank test comparing the frequency of strength training during the different parts of the cycling season.

| Sessions/week | Off-season % | | Pre-season % | | Race season % | |
|---|---|---|---|---|---|---|
| | Male | Female | Male | Female | Male | Female |
| 0 | 19.3 | 18.6 | 13.6 | 1.7 | 34.1 | 18.6 |
| <1 | 5.7 | 3.4 | 1.1 | 1.7 | 29.5 | 15.3 |
| 1 | 4.5 | 10.2 | 15.9 | 10.2 | 28.4 | 45.8 |
| 2 | 52.3 | 47.5 | 61.4 | 66.1 | 6.8 | 18.6 |
| 3 | 17 | 18.6 | 6.8 | 16.9 | 1.1 | 0 |
| 4 | 1.1 | 1.7 | 1.1 | 3.4 | 0 | 1.7 |
| Mean sessions/week | 2.1§ | 2.1§ | 1.9§ | 2.1§ | 0.9 | 1.2 |
| Effect size (r) | .009 | | .260 | | .294 | |
| p | n.s. | | <.01* | | <.01* | |

Mean values excluding cyclists who reported 0 sessions per week, while cyclists who reported <1 times per week counted as 0.5 sessions per week. Abbreviations: § = different from the race season ($p < .01$); n.s. = not significant.

**Table 4. What body parts do you train when strength training, and what type of strength training do you frequently perform.**

| | Male n = 88 | Female n = 59 | Total n = 147 | Sex differences | |
|---|---|---|---|---|---|
| **What body parts do you train?** | n (%) | n (%) | n (%) | V | p-value |
| Upper body | 7 (8)[c] | 16 (27.1)[c] | 23 (15.6)[c] | .259 | <0.01* |
| Core | 58 (65.9)[b] | 44 (74.6)[b] | 102 (69.4)[b] | .092 | n.s. |
| Lower body | 75 (85.2)[a] | 57 (96.6)[a] | 132 (89.8)[a] | .184 | <0.05* |
| **Type of strength training?** | | | | | |
| Maximal strength training (<6 repetitions) | 61 (69.3)[a] | 36 (61)[b] | 97 (66)[a] | .086 | n.s. |
| Hypertrophy training (6–30 repetitions) | 28 (31.8)[b] | 33 (55.9)[b] | 61 (41.5)[b] | .240 | <0.01* |
| Explosive strength training (Lower resistance with maximum movement speed) | 39 (44.3)[b] | 32 (54.2)[b] | 71 (48.3)[b] | .097 | n.s. |
| Core and stability training | 63 (71.6)[a] | 51 (86.4)[a] | 114 (77.6)[a] | .174 | <0.05* |
| Blood flow restriction training | 5 (5.7)[c] | 2 (3.4)[c] | 7 (4.8)[c] | .053 | n.s. |
| Cross fit training | 1 (1.1)[c] | 2 (3.4)[c] | 3 (2)[c] | .078 | n.s. |
| Other forms of strength training | 0 (0)[c] | 2 (3.4)[c] | 2 (1.4)[c] | .143 | n.s. |

Superscript letters indicate significant within-group differences using McNemar's test with Bonferroni correction. Variables that share at least one letter are not significantly different; those without a shared letter are significantly different. Superscripts are only comparable within each column (i.e., within-group). Abbreviations: n = total number of respondents selecting each alternative, % = percentage of respondents, V = Cramer's V, n.s. = not significant.

eventually down to six repetitions. I don't usually go below five repetitions. We stick to the same exercises each time: squats, straight-legged deadlifts for hamstrings and back, weighted calf raises, leg press, and Bulgarian split squats. So those are the five main exercises, along with some simple core exercises. I do three sets per exercise." (CTM3)

Another interviewee described their situation as well:

"They [squats] are quite a cycling-specific movement, I would say. At least in my case, I've performed them with a relatively narrow stance, and the range of motion doesn't go beyond 90 degrees." (PRT4)

Adjustments to strength training practices, usually focused on lessening the load of strength training for the upcoming season, involved reducing the frequency of training, number of exercises, number of sets, number of repetitions, and/or weights used to maintain strength without compromising race performance. The term 'maintenance' was frequently used to describe the focus of strength training during the race season. Training often became less frequent from the pre-season to the race season, with adjustments made to accommodate the race season's demands.

## Challenges with strength training

As Table 5 shows, travelling, fatigue, and racing were the challenges with strength training most reported by the cyclists. Female cyclists most frequently identified travelling as a barrier to strength training, and although the difference was statistically significant (p < 0.01), the effect size was small (V = .235), suggesting only a modest difference between sexes. Similarly, male cyclists more often mentioned a lack of motivation (p < 0.01, V = .242), again with a small effect size. Fatigue was consistently highlighted in interviews as well, especially during the race season due to the demanding schedule and frequent travel. Finding a balance between racing, training, and strength training was difficult, with fatigue often causing strength training to be deprioritised. Travelling also posed logistical challenges by impacting access to suitable facilities. Those findings underscore the complexities of integrating strength training into professional road cycling.

**Table 5. Challenges related to strength training, and perceived negative effects of strength training.**

| | Male n = 88 | Female n = 59 | Total n = 147 | Sex differences | |
|---|---|---|---|---|---|
| **Challenges to performing strength training** | n (%) | n (%) | n (%) | V | p-value |
| Fatigue/soreness affecting the endurance training | 46 (52.3)[a] | 33 (55.9)[a,b] | 79 (53.7)[a] | .036 | n.s. |
| Limited time to perform strength training | 20 (22.7)[b] | 13 (22)[d] | 34 (22.4)[b] | .008 | n.s. |
| Traveling to races and training camps | 49 (55.7)[a] | 44 (74.6)[a] | 93 (63.3)[a] | .235 | <0.01* |
| Restarting strength training after stage races/racing periods | 45 (51.1)[a] | 29 (49.2)[b,c] | 77 (50.3)[a] | .019 | n.s. |
| Lack of knowledge on how to perform strength training | 1 (1.1)[c] | 2 (3.4)[e] | 1 (2)[d] | .078 | n.s. |
| Coach lacking knowledge on how to perform strength training | 0 (0)[c] | 2 (3.4)[e] | 2 (1.4)[d] | .143 | n.s. |
| Lack of interest/motivation to perform strength training | 15 (17)[b] | 1 (1.7)[e] | 16 (10.9)[b,c] | .242 | <0.01* |
| Lack of training equipment/facilities | 18 (20.5)[b] | 14 (23.7)[c,d] | 32 (21.8)[b] | .039 | n.s. |
| Other challenges | 3 (3.4)[c] | 2 (3.4)[e] | 5 (3.4)[c,d] | .034 | n.s. |
| **Negative effects of strength training** | | | | | |
| Impaired cycling performance | 6 (6.8)[d] | 1 (1.7)[c] | 7 (4.8)[d] | .118 | n.s. |
| Soreness impairing endurance training | 64 (72.7)[a] | 38 (64.4)[a] | 102 (69.4)[a] | .088 | n.s. |
| Increased risk of injury | 19 (21.6)[c] | 10 (16.9)[b,c] | 29 (19.7)[c] | .057 | n.s. |
| Increased muscle mass | 42 (47.7)[b] | 24 (40.7)[a,b] | 66 (44.9)[b] | .069 | n.s. |
| No negative effects | 13 (14.8)[c,d] | 11 (18.6)[b,c] | 24 (16.3)[c,d] | .051 | n.s. |

Superscript letters indicate significant within-group differences using McNemar's test with Bonferroni correction. Variables that share at least one letter are not significantly different; those without a shared letter are significantly different. Superscripts are only comparable within each column (i.e., within-group). Abbreviations: n = total number of respondents selecting each, % = percentage of respondents, V = Cramer's V, n.s. = not significant.

"The main challenge is travelling. Suddenly, you can be away and staying in different hotels for 40–50 consecutive days, which makes it difficult to find a gym where you can execute a proper strength programme. Often, the race schedule is unpredictable, with last-minute changes, which makes it impossible to stick to a consistent training plan. If you had planned to do strength training every seventh day, for example, you might not be able to do it because you have to travel to a stage race. It's also hard to plan a visit to a gym to ensure proper strength training. It often turns into an adjusted session or trying to start from scratch each time you resume strength training." (PRT6)

## Rationale for strength training

Cyclists of both sexes cited improved cycling performance, injury prevention, and overall health—particularly bone health—as key reasons for engaging in strength training (Table 6). While some statistically significant sex differences were observed, the associated effect sizes were small. For instance, a greater proportion of female cyclists reported engaging in strength training for injury rehabilitation compared to males (30.5% vs. 15.9%, $p < 0.05$, V = .173). Likewise, improving bone health was cited more frequently by female cyclists (57.6% vs. 40.9%, $p < 0.05$, V = .164). These findings may reflect a slightly greater emphasis on health-related outcomes among female cyclists, but the differences between groups was small. In terms of perceived benefits, female cyclists were also more likely to report improved cycling performance as a result of strength training (93.2% vs. 81.8%, $p < 0.05$, V = .163), although again, the effect size was small. These subtle differences suggest that while core motivations are largely shared across sexes, female cyclists in this sample may perceive slightly greater rehabilitative and performance-related benefits from strength training.

**Table 6. Believed positive effects of strength training, and reasons for why the cyclists perform strength training.**

| Rationale for performing strength training | Male n = 88 | Female n = 59 | | Total n = 147 | Sex differences |
|---|---|---|---|---|---|
| | n (%) | n (%) | n (%) | V | p-value |
| To improve cycling performance | 76 (86.4)[a] | 56 (94.9)[a] | 132 (89.8)[a] | .138 | n.s. |
| To reduce injury risk | 66 (75)[a] | 46 (78)[a] | 112 (76.2)[a] | .034 | n.s. |
| For overall health | 33 (37.5)[b,c] | 25 (42.4)[c,d] | 58 (39.5)[b] | .049 | n.s. |
| To increase muscle mass | 18 (20.5)[b,c,d] | 15 (25.4)[d] | 33 (22.4)[c] | .058 | n.s. |
| Coach/team recommendation | 17 (19.3)[c,d] | 13 (22)[d,e] | 30 (20.4)[c] | .033 | n.s. |
| Rehabilitation from injury | 14 (15.9)[d,e] | 18 (30.5)[c,d] | 32 (21.8)[c] | .173 | <0.05* |
| To improve bone health | 36 (40.9)[b] | 34 (57.6)[b,c] | 70 (47.6)[b] | .164 | <0.05* |
| Other reasons | 3 (3.4)[e] | 2 (3.4)[e] | 5 (3.4)[d] | .001 | n.s. |
| **Believed positive effects of strength training** | | | | | |
| Improved cycling performance | 72 (81.8)[a] | 55 (93.2)[a] | 127 (86.4)[a] | .163 | <0.05* |
| Improved sprinting performance | 62 (70.5)[a,b] | 46 (78)[a,b] | 108 (73.5)[a,b] | .083 | n.s. |
| Reduced risk of injury | 61 (69.3)[a,b] | 46 (78)[a,b] | 107 (72.8)[a,b] | .095 | n.s. |
| Increased muscle mass | 33 (37.5)[c] | 26 (44.1)[c] | 59 (40.1)[c] | .066 | n.s. |
| Improved bone health | 50 (56.8)[b,c] | 37 (62.7)[b,c] | 87 (59.2)[b] | .059 | n.s. |
| Improved overall health | 43 (48.9)[c] | 31 (52.5)[b,c] | 74 (50.3)[c] | .036 | n.s. |
| No positive effects | 0 (0)[d] | 1 (1.7)[d] | 1 (0.7)[d] | .101 | n.s. |

Superscript letters indicate significant within-group differences using McNemar's test with Bonferroni correction. Variables that share at least one letter are not significantly different; those without a shared letter are significantly different. Superscripts are only comparable within each column (i.e., within-group). Abbreviations: n = total number of respondents selecting each, % = percentage of respondents, V = Cramer's V, n.s. = not significant.

"It's for cycling performance and bone health, really. For climbers like me, it's undeniable that there's an impact on bone health, so I try to strengthen as much as possible. And I've actually seen results. Bone density went down when I didn't prioritise strength, and I put a lot of strain on my body, which led to a significant loss of bone mass. However, in the past two years, it's trended upwards again, and I've regained strength." (WTW9)

Most cyclists received coaching guidance, with 90.1% of males and 98.3% of females reported having a coach, while 33.8% of males and 41.4% of females reported having a separate strength coach as well. Statistical analysis revealed no between-sex differences in contentment with coaching for strength ($r = .016$, $p = .844$) or endurance training ($r = .086$, $p = .296$). However, cyclists who engaged in strength training for ≥1 session per week during the race season were more content with their strength coaching ($r = .185$, $p = .025$), and cyclists with a separate strength coach reported greater contentment with their guidance on strength training ($r = .268$, $p = .001$). Although female cyclists reported more enjoyment with strength training than males did ($r = .249$, $p = .002$), no significant sex differences were found in the enjoyment of endurance training ($r = .064$, $p = .438$).

The cyclists with a separate strength coach reported being more content with the guidance that they received on their strength training than ones who did not ($r = .270$, $p = .001$), whereas no differences in contentment of with guidance on endurance emerged ($r = .016$, $p = .837$). They also did not report significantly more enjoyment with either strength training ($r = .155$, $p = .061$) or endurance training ($r = .057$, $p = .488$). Female cyclists reported significantly greater enjoyment with strength training than male cyclists did ($r = .245$, $p = .002$), whereas no difference occurred between males and females in enjoyment with endurance training ($r = .064$, $p = .438$).

Female cyclists (vs. male cyclists; $r = .169$, $p = .04$), cyclists who performed strength training during the race season (vs. ones who did not; $r = .330$, $p = .001$), and cyclists who had a separate strength coach (vs. ones who did not; $r = .260$,

$p$ = .002) were more confident that strength training would improve their cycling performance. Among the cyclists who engaged in strength training, 65.4% of males and 79.6% of females reported noticing improvements in their cycling performance due to strength training.

The interviews revealed varied perceptions of coaching's impact on strength training. Some cyclists praised their coaches for introducing strength training and offering personalised strength training programmes, whereas others emphasised the need for coaches with expertise in cycling as well as in strength and conditioning. Collaboration with national sport institutes also provided specialised guidance. Dedicated strength and conditioning coaches on teams were viewed positively for offering comprehensive support and structured programming. Overall, coaching seemed to play a crucial role in shaping the cyclists' strength training routines, with some expressing satisfaction with their cycling coach's guidance while others preferred specialised strength coaches:

> "I think a problem in the past year, as well with my cycling coach, is that he kind of let me follow my own advice. So I was pretty easy on myself. But I changed teams this year, which means that I now have both a cycling-specific coach and a strength specific coach with the team, and he plans everything." (WTW10)

## Discussion

Our findings showcase the major characteristics of strength training among professional road cyclists. During the off-season and pre-season, a strength training regime of two sessions per week was common among the cyclists in our study. The sessions usually centred on lower body exercises, especially the muscles used in the cycling movements, performed with conventional strength training exercises. Some supplementary core and stability exercises were also performed at the beginning or end of the sessions. Although a large proportion of cyclists stopped strength training during the race season, the cyclists who did not stop usually lowered their frequency of strength training to one session per week while transitioning from the pre-season to the race season. A greater proportion of female than male cyclists performed strength training during the pre- and race season as well as engaged in strength training more frequently during those periods.

The frequency of two strength training sessions per week during the pre-season that we found in our study aligns with recommendations in the literature. For example, Rønnestad and Mujika [5] have argued that two weekly sessions are sufficient for building maximal strength during the pre-season. Among the cyclists in our study, maximal strength training focused on the lower extremities, supplemented with some core training, was the most common method of strength training. Decreasing ranges of repetitions and increasing intensity throughout the period of strength training leading up to the race season seemed to be a common practice among the cyclists. The maximal strength training approach emphasising explosive concentric movement aligns with general recommendations in the literature [3,5,8,9]. Despite its widespread popularity, however, there is a lack of compelling literature supporting the notion that core training positively impacts endurance performance [14]. Such an emphasis on core training might stem from tradition or subjective experiences.

A considerable number of participants in this study—particularly female cyclists—reported engaging in hypertrophy-type strength training. However, the cyclists were quite divided on whether gaining muscle mass was positive or negative, with 40.1% viewing it as a beneficial effect of strength training, while 44.9% considered it a negative one. Although increased muscle mass is often seen as a disadvantage in road cycling due to added weight, there are scenarios in which hypertrophy may be desirable. Sprinters or cyclists with roles requiring high absolute power, rather than power-to-weight ratio, may pursue hypertrophy to enhance force production and sprinting ability. Additionally, some cyclists may include strength training with the aim of preserving muscle mass during periods of high-volume endurance training. It is worth noting, however, that due to the high training volume and energy expenditure typical of professional cycling, substantial muscle hypertrophy is rarely achieved in this population [5].

During the race season, the frequency of strength training among cyclists in our study typically dropped from two sessions per week in the pre-season to one. The reduction is likely adequate to maintain strength adaptations [11]. The sessions were often lower in volume and sometimes also in intensity in order to minimise fatigue and prevent the impairment of subsequent training or racing performance. A significant proportion of cyclists, particularly males, stopped performing strength training during the race season. Only 36.4% of male cyclists engaged in weekly strength training during the race season compared with 66.1% of females. Although strength training during the off-season and pre-season can improve muscle mass and bone health [18], if not maintained then its benefits on cycling performance seem to diminish within 8 weeks [10]. Male cyclists, who compete more frequently than females, have shorter recovery and training times between races, which might contribute to the difference. However, female cyclists identified travelling to races and training camps as a greater challenge, despite competing less often than males. That finding suggests that whereas female cyclists may perceive a host of logistical challenges related to travel, those barriers may not necessarily prevent them from continuing strength training. Female cyclists also reported greater confidence in strength training's effects and more enjoyment with it. A lack of interest or motivation to perform strength training was almost exclusively reported by male cyclists, which might explain some of the disparity in engagement between the sexes during the race season.

For the professional road cyclists in our study, fatigue, travelling, and racing emerged as primary challenges to maintaining consistent strength training, which stands in contrast with amateur long-distance triathletes who cited time restrictions and knowledge as their greatest barriers [33]. Unsurprisingly, our study's cyclists identified time constraints and lack of knowledge challenges less than amateur triathletes have, for professional cyclists are often full-time athletes and can access some of the best expertise and coaching resources available. The challenges of fatigue, travelling, and racing are closely interconnected, however, especially during the race season. Before the race season, fewer competitions and less travel make fatigue from strength training less problematic. However, during the season, frequent competitions make minimising fatigue and interference from strength training crucial, for cyclists need to prioritise recovery in order to achieve peak performance and often neglect strength training in order to avoid extra fatigue and soreness that could impact performance [28,29,31]. Those challenges likely contribute to decreased engagement in strength training during the race season.

Cyclists often face challenges with restarting strength training after long periods of stage racing or frequent racing. Such challenges stem from fatigue due to racing as well as from the reality that the impact of fatigue and muscle soreness from strength training can be greater following periods of minimal exposure to strength training [30]. The cyclists interviewed in our study described difficulties with resuming strength training after such breaks and often experiencing greater soreness and adjusting initial sessions to be less demanding. The demanding racing and travel schedule was a chief reason for some cyclists to avoid strength training during the race season. Added to that, 21.8% of cyclists reported a lack of access to training facilities as a significant barrier, which often led them to postpone strength training or use alternative exercises such as bodyweight workouts and resistance bands, instead of conventional strength training exercises. The issue of lacking training facilities is closely linked to travelling for races or training camps. To address those challenges, teams, coaches, and cyclists may need to develop strategies to integrate strength training into the racing schedule, thereby ensuring consistent engagement despite the obstacles of fatigue, travel, and the unavailability of facilities.

The prospect of improving their cycling performance emerged as the primary rationale driving the cyclists in our study to engage in strength training. Throughout the interviews, the cyclists demonstrated a considerable depth of knowledge and reflection both on their personal strength training regimes and the research on the topic, including both the potential impact on performance and recommended methods. Most cyclists in our study who incorporated strength training reported improvements in their cycling performance as a consequence of strength training. Therefore, it is unsurprising that their decisions about strength training derived from the pursuit of improving performance. Considering the substantial body of research supporting the notion that strength training can benefit endurance and cycling performance [1–6], it is logical to expect cyclists to be well-informed and intentional in leveraging strength training's potential to optimise their performance.

Many cyclists in our study, especially females, focused on strength training's benefits for health, particularly bone health. The cyclists seemed to be aware both of professional cycling's potential drawbacks for bone health [17], especially when trying to keep a low body weight [19,20], and that strength training can positively impact bone health [18]. Therefore, for many cyclists, improving or maintaining bone health was an important reason for them to engage in strength training. During interviews, some cyclists reported regularly testing for bone density and using strength training to improve or maintain their bone health. Engaging in strength training to prevent or recuperate from injuries was also common.

Last, coaching played an important role in shaping cyclists' approaches to strength training. The results of the questionnaire showed that most cyclists had a coach and that a notable proportion had a separate strength coach as well. Those who engaged in strength training during the race season and those with separate strength coaches reported greater contentment with their coaching guidance towards strength training than other cyclists did. In our study, the quantitative data indicate the prevalence of coaching and the cyclists' satisfaction with it, whereas the interviews offered deeper insights into how coaching has influenced decisions about training. Nine out of ten interviewees discussed coaching's role in their strength training while highlighting variations in coaching quality, individualised guidance, and access to specialised strength training programmes. Some cyclists stressed the benefits of working with a dedicated strength coach, whereas others noted a lack of structured strength training guidance from their cycling coaches. Those findings suggest that although coaching can positively influence cyclists' engagement in and perception of strength training, the impact may depend on the coach's expertise and approach. That dynamic highlights the importance of tailored coaching strategies to optimise performance.

## Limitations

Our findings have several limitations. First, they are based on self-reported data collected with a questionnaire and semi-structured interviews. The methods may have introduced response bias and inaccuracies due to participants' subjective recollections of their practices, challenges, and rationales when it comes to strength training. Second, although the sample size of 147 participants and 10 interviews afforded insights, it may not have been sufficiently large or diverse to generalise the findings to all professional cyclists. Third, sampling was biased towards cyclists with links to social media profiles in their rider profiles on FirstCycling.com, who were active on social media and noticed our call for participants, and who were willing to respond to the questionnaire. Fourth, we did not have data on performance indicators such as UCI points, which prevents any direct comparison between the cyclists in our study and the broader population of professional cyclists and may thus limit the generalisability of our findings. Fifth and finally, although the researcher that performed the interviews had a background as a former cyclist and could provide valuable insights into the participating cyclists' perspectives, it could have also inadvertently biased the interview process and thus shaped the formulation and interpretation of data. Those limitations underscore the importance of interpreting our findings cautiously.

## Conclusion

Our study has revealed the practices, challenges, and rationales surrounding strength training among professional road cyclists. Strength training is a common practice in the off-season and pre-season, whereas fewer cyclists engage in strength training during the race season. A typical practice in strength training among the cyclists was approximately two sessions per week during the off-or pre-season and approximately one weekly maintenance session during the race season; those sessions typically focused on maximal strength exercises targeting the lower body, often supplemented by core exercises. Participating female cyclists demonstrated more engagement and more frequent strength training than male cyclists during the pre-season and race season, while challenges such as travelling, fatigue, and racing emerged as significant barriers to maintaining a consistent strength training routine, especially during the race season. Even so, cyclists emphasised the importance of strength training for improving cycling performance, reducing the risk of injury, and promoting bone health. Coaching guidance played a crucial role in enhancing cyclists' confidence and satisfaction with their strength training regimen, especially when provided by separate strength coaches.

## Supporting information

**S1 Fig.  Survey: Strength training and cycling performance.**
(DOCX)

## Author contributions

**Conceptualization:** Vidar Vikestad, Idar Kristian Lyngstad, Terje Dalen.

**Data curation:** Vidar Vikestad, Terje Dalen.

**Formal analysis:** Vidar Vikestad, Idar Kristian Lyngstad, Terje Dalen.

**Investigation:** Vidar Vikestad, Terje Dalen.

**Methodology:** Vidar Vikestad, Idar Kristian Lyngstad, Terje Dalen.

**Project administration:** Terje Dalen.

**Resources:** Vidar Vikestad.

**Software:** Vidar Vikestad.

**Supervision:** Idar Kristian Lyngstad, Terje Dalen.

**Validation:** Vidar Vikestad, Terje Dalen.

**Visualization:** Vidar Vikestad, Terje Dalen.

**Writing – original draft:** Vidar Vikestad.

**Writing – review & editing:** Vidar Vikestad, Idar Kristian Lyngstad, Terje Dalen.

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
