## [Decision Letter · Decision Letter 0]

Dear Dr. Dalen,

A review and revision should be performed by a native English speaker to clarify language, syntax, and grammar usage.

Clearly define, perhaps in a supplement different types of strength training and keep these distinct throughout the paper. Avoid mashing up the distinctions, as it gets confusing reading the paper; eg, strength and hypertrophy training generally result in similar adaptations, while specific “core” must be specifically defined and cannot be used synonymously with “stability” training, which should be defined. Moreover, how the “core” is defined muddies this.  Similarly, clearly define coach (sport vs S&C). Endurance coaches can and often do fill a strength coach role, but a “strength coach” or as distinguished by groups like the NSCA, the strength and conditioning coach, is a unique role.The authors should, as noted by reviewer 2, move the Informed consent/ethics section to near the beginning to make it clear that was performed first.The selection of participants to the sample is self-selection.  While the authors do not explicitly estimate treatment effects or draw strong conclusions about the population of UCI pro cyclists, there is a risk that the sample is significantly different based on self-selection.  The authors should do some analysis of this, such as the % of male vs female respondents vs % contacted.  Compare the mix or WT vs Conti tour; can they find the average age and UCI points of each type of rider and compare the population to the sample? If this cannot be done some effort or discussion and admission of limitations/potential bias is needed.Secondarily, I note that there were just two women’s teams participating. Was there a team bias? IME across sports, the participation of strength training among endurance athletes in particular takes on a philosophical or dogmatic position, particularly among women. Do we know what the general practices/advice are among other teams? If not, tt should be noted that not all teams adhere to similar training practices and this too may be a bias.Both reviewers 1 and 2 ask for clarity in the methods, so please take head of this.Please carefully consider the recommendations on results presented by reviewer 1, as well as the notes made by 1 & 2 on statistical aspects of the paper.

We look forward to receiving your revised manuscript.

Kind regards,

Chris Harnish, PhD

Academic Editor

PLOS ONE

Reviewers' comments:

Reviewer's Responses to Questions

**Comments to the Author**

1. Is the manuscript technically sound, and do the data support the conclusions?

Reviewer #1: Partly

Reviewer #2: Yes

Reviewer #3: Yes

2. Has the statistical analysis been performed appropriately and rigorously?

Reviewer #1: Yes

Reviewer #2: Yes

Reviewer #3: Yes

3. Have the authors made all data underlying the findings in their manuscript fully available?

Reviewer #1: Yes

Reviewer #2: Yes

Reviewer #3: Yes

4. Is the manuscript presented in an intelligible fashion and written in standard English?

Reviewer #1: Yes

Reviewer #2: Yes

Reviewer #3: Yes

Reviewer #1: Overall, this is a good start at a paper. It is a good paper, with well written English, with good data but the authors fail to make use of the good data they have; a number of small errors and lack of clarity also are present. The attached document gives all my comments in detail.

Reviewer #2: Ref #: PONE-D-24-55968.

TITLE: Professional UCI Road cyclists’ practices, challenges and rationales towards strength training.

SUMMARY: This study examines the strength training practices of professional UCI road cyclists. This is a topical study that outlines challenges and views of some professional road cyclists to strength training and represents an important area of study within the fields of exercise science and strength and conditioning. The introduction is reasonably well written and clarifies some need for the study. The methods section lacks clarity in key areas and greater detail should be provided by the authors to aid replication, for instance; How many cyclists were contacted? Were all cyclists on the FirstCycling.com database contacted and if not, how did the authors arrive at the specific cyclists to be contacted? How many cyclists responded and what was the response rate? Were all cyclists selected for interview? How did the authors select cyclists for interview, specifically was this a random or convenience sample? The Methods section overall would be enhanced by presenting with more traditional Participants, Procedures and Data / Statistical Analysis sections. The Statistical Analysis section should clarify how the authors treated data and how they are interpreting specific statistics – e.g. Z-scores. The Results section is well presented and written. The authors are urged to refer to interview data in a less general manner since interview data are taken from 10 cyclists and not all the cyclists surveyed or all cyclists. The Discussion section is very well written, and Conclusions made are appropriate.

ABSTRACT; This is a well written abstract. The authors should consider the following changes:

Line 12; How many professional road cyclists were contacted? This should be clarified for the reader.

INTRODUCTION: Line 39; In the first paragraph the authors discuss the general benefits of strength training for cyclists / endurance athletes. At the beginning of the second paragraph, they refer to core and stability strength training. It would aid clarity for the reader if the authors defined the meaning of core and stability strength training and outlined how this is different from strength training discussed in the first paragraph. This is particularly important since the NSCA define core exercises as those involving recruitment of one or more large muscle areas involving two or more primary joints (multijoint exercises), and that receive priority when one is selecting exercises because of their direct application to the sport. Does the author’s definition of core and stability strength training align with this, or are they referring to a different type of training?

Line 40; The authors discuss findings of Sitko et al. Greater clarity is required here. What is meant by power across durations of 5 seconds to 20 minutes? How was power assessed?

Line 41; The authors should provide some examples to clarify their meaning of core and classical strength training exercises.

Line 48; What is meant by bone markers? Are the authors referring to bone mineral density?

Line 73; The authors refer to previous work looking at the training practices of Tour de France finishers. Did the authors of this work clarify reasons for strength training not continuing six months prior to the event? This should be explicitly stated for the reader to provide greater rationale for the current work.

METHODS: Line 87; How many cyclists were contacted? Were all cyclists on the FirstCycling.com database contacted and if not, how did the authors arrive at the specific cyclists to be contacted? How many cyclists responded and what was the response rate? Were all cyclists selected for interview? How did the authors select cyclists for interview, specifically was this a random or convenience sample?

Line 108; Insert ‘being’ after before.

Line 117; The authors discuss collection of session frequency data collected via questionnaire and provide a scale of 0, <1, and 1 – 7 sessions per week. What is meant by less than 1 if 0 is provided on the scale? Is it assumed that each session lasts approximately 1-hour? Are there any questions about the duration of each strength training session?

Line 142; Were all interviews conducted by the same person? How long did each interview last for each participant?

Line 161; The authors should move the section on Informed Consent and Ethical Practices to the end of the Participants section to clarify to the reader that informed consent and ethical approval was achieved.

RESULTS: Line 184; The authors state that a large percentage of cyclists ceased to perform strength training from pre-season to race season. This is repetitive (stated in line 180 also) and suggests that a large percentage ceased to perform strength training during race season. The authors should include the specific percentage in parentheses here also.

Table 3; Z-scores are presented presumably looking at the magnitude of differences between male and female cyclists. Details of how z-scores were interpreted should be clarified in the Methods section.

Line 200; The authors suggest that strength training practices “typically” comprised two weekly sessions. They should consider rewording to clarify that the athletes interviewed typically completed two sessions per week. The authors should clarify the specific season they are referring to since data is presented based on the training phase.

Line 259; The authors report data suggesting the availability of a coach? Is this a strength and conditioning coach or a sports coach knowledgeable about strength training?

Line 268; Is it meant that the “riders” that had a separate strength coach?

Line 275; Check wording of this sentence.

Line 282; Change “strength conditioning” to strength and conditioning.

DISCUSSION: Line 305; The authors should name the author for readability – i.e. Rønnestad and Mujika.

Line 307; Change “in this study” to “in the current study”. This will aid clarity and create distinction from the previous study cited.

Line 342; The authors should refer to the cyclists interviewed rather than many cyclists.

Line 363; Check wording at the end of this sentence.

Reviewer #3: A good wee study, with interest, and I think will draw readers in from a coaching and strength and conditioning background. I look forward to seeing the article in publication, and expect it will be cited in future work by cycling and strength and conditioning researchers.

**Do you want your identity to be public for this peer review?** For information about this choice, including consent withdrawal, please see our Privacy Policy

Reviewer #1: **Yes: ** Dr Gregory P. Swinand

Reviewer #2: No

Reviewer #3: No

---

## [Author Response · Author response to Decision Letter 1]

20 Feb 2025

Response to Editor and Reviewers

Dear all:

Thank you very much for your comments and the opportunities to improve our manuscript. We have carefully reviewed each of your comments and tried to make changes according to your comments. Thank you very much for your constructive contributions; we believe this has improved the quality of the article and hope that the attached version will be suitable for publication.

Editor:

Comment 1: A review and revision should be performed by a native English speaker to clarify language, syntax, and grammar usage.

• Response 1: We acknowledge this suggestion and will ensure a thorough language revision by a native English speaker before final submission.

Comment 2: Clearly define, perhaps in a supplement different types of strength training and keep these distinct throughout the paper. Avoid mashing up the distinctions, as it gets confusing reading the paper; eg, strength and hypertrophy training generally result in similar adaptations, while specific “core” must be specifically defined and cannot be used synonymously with “stability” training, which should be defined. Moreover, how the “core” is defined muddies this. Similarly, clearly define coach (sport vs S&C). Endurance coaches can and often do fill a strength coach role, but a “strength coach” or as distinguished by groups like the NSCA, the strength and conditioning coach, is a unique role.

• Response 2: In the questionnaire, "core and stability training" were combined into one category; therefore, separating them in the results is not feasible. However, we have revised the manuscript to ensure that distinctions between strength training types are clearer throughout the text, and we provide a definition of "core and stability training" in the introduction. Additionally, we have clarified the distinction between endurance coaches and strength & conditioning (S&C) coaches.

Comment 3: The authors should, as noted by reviewer 2, move the Informed consent/ethics section to near the beginning to make it clear that was performed first.

• Response 3: Revised as suggested.

Comment 4: The selection of participants to the sample is self-selection. While the authors do not explicitly estimate treatment effects or draw strong conclusions about the population of UCI pro cyclists, there is a risk that the sample is significantly different based on self-selection. The authors should do some analysis of this, such as the % of male vs female respondents vs % contacted. Compare the mix or WT vs Conti tour; can they find the average age and UCI points of each type of rider and compare the population to the sample? If this cannot be done some effort or discussion and admission of limitations/potential bias is needed.

• Response 4: The ”Participants” section is revised based on the comments of editor and the reviewers for a more details and better clarification of the recruitment process. We have also revised the “Limitations” section so that it addresses these concerns. We do not have data to compare UCI points, but we found the number of riders in the whole population. We do not see it as necessary to include this in the study, but let us know if you think it should be included.

We also found the total number of cyclists for each category:

536 WT

404 PRT

1413 CTM (European teams)

240 WTW

667 CTW (European teams)

Comment 5: Secondarily, I note that there were just two women’s teams participating. Was there a team bias? IME across sports, the participation of strength training among endurance athletes in particular takes on a philosophical or dogmatic position, particularly among women. Do we know what the general practices/advice are among other teams? If not, it should be noted that not all teams adhere to similar training practices and this too may be a bias.

• Response 5: We would like to clarify that the study included respondents from two levels of women’s teams (Women’s World Teams and Women’s Continental Teams), not just two individual teams. We will revise the phrasing in the methods section to make this distinction clearer.

Comment 6: Both reviewers 1 and 2 ask for clarity in the methods, so please take head of this.

• Response 6: We have made our best efforts to revise the method based on the suggestions from both reviewers.

Comment 7: Please carefully consider the recommendations on results presented by reviewer 1, as well as the notes made by 1 & 2 on statistical aspects of the paper.

• Response 7: We have made changes to the results and the statistics as a result of the suggestions from the reviewers. This includes within-group analysis and the use of effect sizes and how these are interpreted.

Reviewer #1:

Comment 1: The biggest criticism of the paper is that the authors fail to make good use of what would appear to be good data and overall decent method. For example, the results of the scale and questionnaire –how is this incorporated? While table 2 gives quite a detailed list of items covered, the reader is then expecting to see how these items correlated or impacted the overall outcomes in terms of strength training; yet this is missing.

• Response 1: We appreciate the concern regarding the incorporation of the questionnaire and interview data into the results section. We would like to clarify that the key themes from Table 2 are thoroughly covered throughout the Results section, with findings supported by both quantitative data (e.g., statistical analyses of training frequency, challenges, and rationales) and qualitative insights from the most meaningful interviews.

For instance:

• Challenges related to strength training (e.g., fatigue, traveling, and race schedules) are detailed in Table 5 and further elaborated with direct interview quotes, such as a rider describing the difficulties of maintaining strength training while traveling for long race periods:

"Suddenly, you can be away and staying in different hotels for 40-50 consecutive days, making it difficult to find a gym where you can execute a proper strength program." (6PRT)

• Adjustments to strength training across seasons are presented in Table 3 and expanded upon in the interviews, where cyclists described their periodization approach:

"We start with a conditioning phase with 10-12 repetitions, then gradually decrease after 4-5 weeks, down to 8 repetitions, and eventually down to 6 repetitions." (3CTM)

• Rationale for strength training choices, including its perceived benefits for cycling performance, injury prevention, and bone health, is presented in Table 6 and supported by qualitative insights. For example, one cyclist highlighted the impact of strength training on bone density:

"As a climber, it's undeniable that there's an impact on bone health, so I try to strengthen it as much as possible. And I have actually seen results." (9WTW)

We believe that by integrating statistical findings with direct participant perspectives, the study presents a comprehensive and nuanced understanding of the data. However, we acknowledge the importance of making these connections more explicit and will consider minor adjustments in the Discussion section to further reinforce how the themes from Table 2 are represented in the results.

Comment 2: Getting to table 3, we find that, perhaps unsurprisingly, cyclists did more strength training in the pre season than in the off season, and less still during the race season.

i. Table 3 is really only comparing male vs female in terms of number of sessions/week; we find a very small difference between sexes in pre and during the season, but not in the offseason. While statistically significant, the order of the difference is 0.2x per week. Is this important? Does it inform coaches and amateur cyclists? Professional cyclists?

1. Could this difference be driven by the fact that male cyclists likely race more days, race longer distances, races have more climbing, have more travel, and are perhaps more specialised in team role (e.g., sprinter, climber, etc?). The reader is left wondering.

• Response 2: We acknowledge that while the sex differences in strength training frequency during the pre-season and race season are statistically significant, their practical relevance requires further consideration. To better interpret the magnitude of these differences, we have replaced Z-scores with effect sizes in the results. Additionally, we address potential explanations for these differences, such as race days and traveling, in the discussion section rather than the results.

Comment 3: Table 4 again shows the types of strength training, but the statistics show only the statistical differences between male and females.

ii. Again the issue of core, vs upper and lower body and strength is confusing.

iii. The categorisation of ‘explosive’ and ‘maximal’ – does this make sense? the overall categorisation has several categories for which the insignificance of the difference results are driven by the low numbers. It would make sense to group these and test the difference overall for different grouped categories.

• Response 3: We have now included within-group statistics to asses the magnitude of the various variables. Regarding the classification of maximal and explosive strength training, we have maintained these as separate categories since they differ in execution and intent. Maximal strength training was defined by a low repetition range (<6 reps), whereas explosive strength training was characterized by lighter loads and an emphasis on maximal movement speed. Given these distinct training methodologies, we believe keeping them separate provides a more accurate reflection of training practices.

Comment 4: Table 5 shows the challenges to strength training but we are only testing the statistical difference between males and females;

iv. perhaps surprisingly, the only things that turn out significant is the travel time to races and camps.

v. The lack of interest seems to be driven by a small number in the females.

vi. A number of interesting details of this table are not discussed – for example:

1. Lack of coaches or facilities is not driving differences between males and females.

2. Surprising muscle mass gain negative is not bigger for males.

3. Almost 36% of females vs 27% of males say soreness is a negative, yet this difference is not statistically significant?

• Response 4: We have now included within-group statistics to asses the magnitude of the various variables.

Comment 5: I don’t understand the differences in the two results for the top part of the table on ‘Fatigue/soreness affecting the endurance training’ and ‘Soreness impairing endurance training’—how did the same question give such different results?

• Response 5: Thank you for noticing this mistake! We had plotted the wrong number, but the mistake is now corrected.

Comment 6: What table 5 really needs is analysis both across and down the columns; how do the different things stack up against each other?

Then these results should be analysed including the other covariates the authors say they collected? What about UCI rank? What about age? Years prof? What about whether the rider is climber, sprinter or TT or Gregario/Domestique?

• Response 6: We have now included within-group analysis. We agree that this would be interesting to include data of UCI rank and other rider data, but this is unfortunately not included in this set of data and should be taken into account when interpreting the results.

Comment 7: Table 6 then the critique is more or less the same?

vii. We only see stat test of diff between sexes but so what?

viii. Weak stat diff but no analysis down the table

ix. We see for example ‘to improve sprinting performance’ but was the sample matched male/female in terms of how many sprinters vs climbers were male vs female?

x. Above all, we have the rationale for why they did it and then whether they think it worked – how did the matched sample compare across these? (did people achieve their goals in terms of perceived benefits?).

• Response 7: We have now included within-group analysis. Regarding the type of cyclists in each group. We did not ask them that type of cyclists they consider themselves, but we did ask them what duration of effort they would consider themselves the strongest. This included everything from short sprint durations to durations of over one hour. There were no differences between the groups for any of the durations as stated in the beginning of the results section. Regarding perceived benefits, in the second to last paragraph of the results we state that “Among the cyclists performing strength training, 65.4 % of male cyclists, and 79.6 % of female cyclists reported that they had noticed improvements in cycling performance from the strength training.”

Comment 8: There are added discussions which leave the reader wondering.

Starting at lines 259, they do some analysis of how content folks were with their coaching, etc, but did coaching guidance have a benefit and help achieve success in terms of translating the perceived benefits into perceived success a la table 6?

• Response 8: In the second-to-last paragraph of the results section, we report that 65.4% of male and 79.6% of female cyclists who performed strength training perceived improvements in their cycling performance. While this suggests a link between strength training and perceived success, the study did not specifically analyze whether coaching guidance directly influenced these perceived benefits.

Comment 9: Starting at 275, they state:

“275 Both female vs. male (Z= -2.051, p=0.04), those who trained strength during the racing season vs. those 276 who did not (Z= -4.001, p=0.001), and those who had a separate strength coach vs. those who did not (Z= 277 -3.148, p=0.002), was more confident that strength training would improve their cycling performance.” But this isn’t really telling us anything as no-doubt the riders who thought it ‘would work’ did more of it; but how did it turn out?

• Response 9: We acknowledge that confidence in strength training may influence engagement rather than directly indicating effectiveness. However, 65.4% of male and 79.6% of female cyclists who engaged in strength training reported perceived performance improvements. While this suggests a link between training and benefits, we recognize the limitation of self-reported data. Future research should assess objective performance measures to confirm these effects.

Comment 10: Some of the conclusions don’t seem to be adequately supported or contrast with the actual results, for example:

b. ‘Coaching significantly influenced cyclists' strength training approaches. Most cyclists received 371 coaching, with many having a separate strength coach.’

i. Does it really show this? Did we see the choices of types of strength training table 4 broken down by with/without coaching, or did they regress a multichotomous dependent variable on factors including a dummy variable for coach/no coach?

• Response 10: We acknowledge that our original phrasing may have overstated the statistical evidence supporting the influence of coaching on strength training approaches. The questionnaire provided a quantitative overview of coaching prevalence and satisfaction, but it did not directly assess whether coaching impacted specific strength training choices (e.g., exercise selection in Table 4). However, the interviews offered additional context, as nine out of ten interviewees discussed the role of coaching in shaping their strength training practices. To clarify this, we have revised the discussion to put greater emphasis on the qualitative insights while ensuring that conclusions align with the presented data.

Comment 11: “Male cyclists, who 323 compete more frequently, have shorter recovery and training times between races, that might contribute 324 to this difference.”

ii. Surprisingly, perhaps, the difference in table 5 is that a significantly higher % of women (c. 75%) say this is why they don’t do/face challenges to do strength training;

• Response 11: We have revised the discussion to better reflect the data regarding the differences in strength training engagement between male and female cyclists.

Comment 12: The selection of participants to the s

---

## [Decision Letter · Decision Letter 1]

Dear Dr. Dalen,

**Clarify your use of superscripts using either captions or foot notes to articulate data interpretation and when multiple superscripts are used.****Provide a bit more interpretive text regarding statistically significant data.****Avoid emotive terms in the text.****Provide rationale for when road cyclists might desire hypertrophy; this last point is important, historically "bigger" or heavier is seen as negative to performance.**

We look forward to receiving your revised manuscript.

Kind regards,

Chris Harnish, PhD

Academic Editor

PLOS ONE

Journal Requirements:

Reviewers' comments:

Reviewer's Responses to Questions

**Comments to the Author**

Reviewer #1: All comments have been addressed

Reviewer #2: All comments have been addressed

2. Is the manuscript technically sound, and do the data support the conclusions?

Reviewer #1: Yes

Reviewer #2: Yes

3. Has the statistical analysis been performed appropriately and rigorously?

Reviewer #1: Yes

Reviewer #2: Yes

4. Have the authors made all data underlying the findings in their manuscript fully available?

Reviewer #1: Yes

Reviewer #2: Yes

5. Is the manuscript presented in an intelligible fashion and written in standard English?

Reviewer #1: Yes

Reviewer #2: Yes

Reviewer #1: The authors have done a good job addressing the comments. I have three small comments. 1) the presentation of the statistical results and differences with the superscripts is a good way, but in the caption or a FN spell out more that this is interpretable both down and across. (if i've understood correctly). What does it mean when there are more than one superscripts? Finally, some added interpretation of the results where things are stat signficant. 2) give a good once over again, avoid emotive terms such as 'whopping'. 3) for just road cyclists -why is hypertrophy ever 'good' or the goal?

Reviewer #2: SUMMARY: This study examines the strength training practices of professional UCI road cyclists. This is a topical study that outlines challenges and views of some professional road cyclists to strength training and represents an important area of study within the fields of exercise science and strength and conditioning. The authors have addressed my concerns. The manuscript reads well and is clear. Please see the minor grammatical / typographical errors highlighted below.

INTRODUCTION: Line 60; typographical error. Check spelling of hypertrophy.

RESULTS: Line 316; check wording of strengthening coaches here. Since strengthening is a verb, the authors are urged to refer to the strength (or strength and conditioning) coach. They should check the manuscript to ensure that they are referring to the strength (or strength and conditioning) coach consistently throughout – e.g. see Line 444 also.

**Do you want your identity to be public for this peer review?** For information about this choice, including consent withdrawal, please see our Privacy Policy

Reviewer #1: **Yes: ** Dr Gregory P. Swinand

Reviewer #2: No

---

## [Author Response · Author response to Decision Letter 2]

5 May 2025

Peer review r2:

Editor comments:

Comment 1: Clarify your use of superscripts using either captions or foot notes to articulate data interpretation and when multiple superscripts are used.

Response 1: We have revised the captions for Tables 4–6 to clearly explain the use of superscript letters. These indicate within-group comparisons; values sharing a letter are not significantly different, while those with different letters are. When multiple superscripts appear, the variable overlaps statistically with multiple others.

Example: If one value is marked ᵃ, another is ᵇ, and a third is ᵃᵇ, the ᵃᵇ value is statistically similar to both the ᵃ and ᵇ values, but the ᵃ and ᵇ values differ significantly from each other.

Comment 2: Provide a bit more interpretive text regarding statistically significant data.

Response 2: We have revised the results section to include more interpretation of statistically significant results, focusing on what these differences mean practically for strength training among professional cyclists.

Comment 3: Avoid emotive terms in the text.

Response 3: We have reviewed the manuscript and removed or revised emotive language such as “whopping” to ensure a neutral and academic tone throughout.

Comment 4: Provide rationale for when road cyclists might desire hypertrophy; this last point is important, historically "bigger" or heavier is seen as negative to performance.

Response 4: We have expanded the discussion (L348-357) to provide additional rationale for why some road cyclists may pursue hypertrophy. While excessive muscle mass is generally seen as counterproductive in endurance sports, hypertrophy may be beneficial in certain cases — such as for sprinters or cyclists focused on power output. Additionally, some athletes may aim to maintain or prevent loss of muscle mass. We also acknowledge that due to the high training loads typical in professional cycling, substantial hypertrophy is often difficult to achieve, and gains tend to be modest.

Reviewer #1:

Comment 1: The presentation of the statistical results and differences with the superscripts is a good way, but in the caption or a FN spell out more that this is interpretable both down and across. (if i've understood correctly). What does it mean when there are more than one superscripts? Finally, some added interpretation of the results where things are stat significant.

Response 1: We have revised the captions for Tables 4–6 to clearly explain the use of superscript letters. These indicate within-group comparisons; values sharing a letter are not significantly different, while those with different letters are. When multiple superscripts appear, the variable overlaps statistically with multiple others.

Example: If one value is marked ᵃ, another is ᵇ, and a third is ᵃᵇ, the ᵃᵇ value is statistically similar to both the ᵃ and ᵇ values, but the ᵃ and ᵇ values differ significantly from each other.

We have revised the results section to include more interpretation of statistically significant results, focusing on what these differences mean practically for strength training among professional cyclists.

Comment 2: Give a good once over again, avoid emotive terms such as 'whopping'.

Response 3: We have reviewed the manuscript and removed or revised emotive language such as “whopping” to ensure a neutral and academic tone throughout.

Comment 3: for just road cyclists -why is hypertrophy ever 'good' or the goal?

Response 3: Response 4: We have expanded the discussion (L348-357) to provide additional rationale for why some road cyclists may pursue hypertrophy. While excessive muscle mass is generally seen as counterproductive in endurance sports, hypertrophy may be beneficial in certain cases — such as for sprinters or cyclists focused on power output. Additionally, some athletes may aim to maintain or prevent loss of muscle mass. We also acknowledge that due to the high training loads typical in professional cycling, substantial hypertrophy is often difficult to achieve, and gains tend to be modest.

Reviewer #2:

Comment 1: INTRODUCTION: Line 60; typographical error. Check spelling of hypertrophy.

Response 1: Corrected the spelling error in ‘hypertrophy’.

Comment 2: RESULTS: Line 316; check wording of strengthening coaches here. Since strengthening is a verb, the authors are urged to refer to the strength (or strength and conditioning) coach. They should check the manuscript to ensure that they are referring to the strength (or strength and conditioning) coach consistently throughout – e.g. see Line 444 also.

Response 2: We have reviewed the manuscript and replaced all instances of “strengthening coach” with the correct term “strength coach” for clarity and consistency.

---

## [Editor Report · Decision Letter 2]

Strength training among professional UCI road cyclists: Practices, challenges, and rationales

PONE-D-24-55968R2

Dear Dr. Dalen,

We’re pleased to inform you that your manuscript has been judged scientifically suitable for publication and will be formally accepted for publication once it meets all outstanding technical requirements.

Kind regards,

Domingo Jesús Ramos-Campo, Ph.D

Academic Editor

PLOS ONE

Additional Editor Comments (optional):

The article is ready to be published
---

## [Editor Report · Acceptance letter]

PONE-D-24-55968R2

PLOS ONE

Dear Dr. Dalen,

I'm pleased to inform you that your manuscript has been deemed suitable for publication in PLOS ONE. Congratulations! Your manuscript is now being handed over to our production team.

Kind regards,

on behalf of

Dr. Domingo Jesús Ramos-Campo

Academic Editor

PLOS ONE